# INFORMED WEIGHT INITIALIZATION OF GRAPH NEURAL NETWORKS AND ITS EFFECT ON OVERSMOOTHING

## ABSTRACT

In this work, we generalize the ideas of Kaiming initialization to Graph Neural Networks (GNNs) and propose a new initialization scheme that addresses the problem of oversmoothing. GNNs are typically initialized using methods, that have been designed for other types of Neural Networks, such as Xavier or Kaiming initialization. Such methods ignore the underlying topology of the graph. We propose a new initialization method, called G-Init, which takes into account (a) the variance of signals flowing forward, (b) the gradients flowing backward through the network, and (c) the effect of graph convolution, which tends to smooth node representations and leads to the problem of oversmoothing. Oversmoothing is an inherent problem of GNNs, which appears when their depth increases, making node representations indistinguishable. We show that in deep GNNs, G-Init reduces oversmoothing and enables deep architectures. We also verify the theoretical results experimentally.

## 1 INTRODUCTION

Weight initialization has been shown to play an important role in the training of neural networks (He et al., 2015; Glorot & Bengio, 2010). Choosing in an informed manner the initial weight values can significantly impact the training convergence and the final performance of the model. Informed weight initialization methods aim to balance between avoiding convergence to suboptimal solutions and preventing exploding or vanishing gradients. The most prominent techniques take into account the network's architecture and the activation functions, aiming to stabilize the variance of signals flowing forward and gradients flowing backward through the model.

Despite their success in Feed Forward Networks (FFNs), the aforementioned weight initialization methods are not directly applicable to GNNs. The underlying graph structure and the message passing in GNNs affects the information flow between network neurons, which in turn impacts the variance of the signals flowing forward and the gradients flowing backward. Additionally, when the architecture gets deeper, the variance between node representations tends to decrease, due to oversmoothing.

Meanwhile, the interest in deep Graph Neural Networks (GNNs) has been increasing (Chen et al., 2020; Rong et al., 2020), but common GNN models, like the Graph Convolutional Network (GCN) quickly lose their ability to create informative node representations as their depth increases (Li et al., 2018). This is due to the neighborhood aggregation occurring in every layer of the model, which is a type of Laplacian smoothing. Stacking multiple layers in GNNs leads to the over-smoothing phenomenon (Oono & Suzuki, 2020; Cai & Wang, 2020), where node representations are indistinguishable, which in turn harms model's performance.

Several methods have been proposed to alleviate oversmoothing and enable deep GNNs (Zhou et al., 2021; Zhao & Akoglu, 2020), but none of them considers the effect of weight initialization. The aim of this paper is primarily to propose a weight initialization that is suitable for GNNs and secondarily to investigate the effect of weight initialization to oversmoothing. In particular, we generalize the analysis of He et al. (2015) to GNNs and present results about the variance of signals and gradients flowing inside the network. Utilizing these theoretical results, we propose a novel weight initialization method (G-Init), that stabilizes variance. Furthermore, we show the effect of G-Init to the largest singular values of the weight matrices, which in turn reduces the oversmoothing

effect (Oono & Suzuki, 2020). Finally, experiments on 8 datasets verify our theoretical results. Hence, the main contributions of this paper are as follows:

- **Theoretical analysis of weight initialization for GNNs:** We generalize the analysis of He et al. (2015) to GNNs and derive the respective formulas about the variance of the forward signals and the backward gradients within the model.

- **A new weight initialization method (G-Init):** We propose a new way to initialize GNN weight matrices, which stabilizes the variance, and show its relationship to oversmoothing.

- **Deep GNNs:** We experiment with deep GNNs up to 64 layers across 8 datasets and show that the proposed initialization reduces oversmoothing.

## 2 NOTATIONS AND PRELIMINARIES

### 2.1 NOTATIONS

We will focus on the common task of semi-supervised node classification on a graph. The graph under investigation is $G(\mathbb{V}, \mathbb{E}, X)$, with $|\mathbb{V}| = N$ nodes $u_i \in \mathbb{V}$, edges $(u_i, u_j) \in \mathbb{E}$ and $X = [x_1, ..., x_N]^T \in \mathbb{R}^{N \times C}$ the initial node features. The edges form an adjacency matrix $A \in \mathbb{R}^{N \times N}$ where edge $(u_i, u_j)$ is associated with element $A_{i,j}$. $A_{i,j}$ can take arbitrary real values indicating the weight (strength) of edge $(u_i, u_j)$. Node degrees are represented through a diagonal matrix $D \in \mathbb{R}^{N \times N}$, where each element $d_i$ represents the sum of edge weights connected to node i. During training, only the labels of a subset $V_l \in \mathbb{V}$ are available. The task is to learn a node classifier, that predicts the label of each node using the graph topology and the given feature vectors.

**GCN** originally proposed by Kipf & Welling (2017), utilizes a feed forward propagation as:

$$H^{(l+1)} = \sigma(\hat{A} H^{(l)} W^{(l)}) \tag{1}$$

where $H^{(l)} = [h_1^{(l)}, ..., h_N^{(l)}]$ are node representations (or hidden vectors or embeddings) of the $l$-th layer, with $h_i^{(l)}$ standing for the hidden representation of node i; $\hat{A} = \hat{D}^{-1/2}(A + I)\hat{D}^{-1/2}$ denotes the augmented symmetrically normalized adjacency matrix after self-loop addition, where $\hat{D}$ corresponds to the degree matrix; $\sigma(\cdot)$ is a nonlinear element-wise function, i.e. the activation function, which is typically ReLU; and $W^{(l)}$ is the trainable weight matrix of the $l$-th layer.

### 2.2 UNDERSTANDING OVERSMOOTHING

Li et al. (2018) showed that graph convolution is a type of Laplacian smoothing. Utilizing that smoothing process, the model creates similar node representations within each (graph) cluster, i.e. densely connected group of nodes, which in turn improves the performance on semi-supervised tasks on graphs. Increasing the depth of the model leads to repetition of the smoothing operation multiple times and ultimately to oversmoothing of node representations, i.e. node representations become similar and a fraction of the initial information is lost.

Oono & Suzuki (2020) have generalized the idea in Li et al. (2018) considering also, that the ReLU activation function maps to a positive cone. They explain oversmoothing as a convergence to a subspace, and provide an estimate of the speed of convergence to this subspace. That speed is expressed as the distance of node representations from the oversmoothing subspace $M$ (details can be found in Oono & Suzuki (2020)).

**Theorem 1 (Oono & Suzuki (2020))** *Let* $s_l = \prod_{h=1}^{H_l} s_{lh}$ *where* $s_{lh}$ *is the largest singular value of weight matrix* $W_{lh}$, $s = sup_{l \in N^+} s_l$. *then* $d_M(X^{(l)}) = O((s\lambda)^l)$, *where l is the layer number and if* $s\lambda < 1$ *the distance from oversmoothing subspace exponentially approaches zero. Where* $\lambda$ *is the smallest non-zero eigenvalue of* $I - \hat{A}$.

Based on Theorem 1, deep GCNs are expected to suffer from oversmoothing. Since the topology is predefined, to reduce oversmoothing the model would have to maintain a high value of the product of the largest singular values of the weight matrices.

### 2.3 WEIGHT INITIALIZATION

Before training, all entries of the weight matrices are sampled from a probability distribution. Selecting the appropriate distribution is of high importance. The most notable methods (i.e., Glorot & Bengio (2010) and He et al. (2015)) focus on the stabilization of the variance across layers. They aim to stabilize both the variance of the signals flowing forward and of the gradients which flow backwards. The aforementioned methods either use uniform or zero-mean Gaussian distributions. When Gaussian distributions are used, their variance plays crucial role and constitutes the essence of the analysis conducted by Glorot & Bengio (2010); He et al. (2015). In particular, Glorot & Bengio (2010) proposed to initialize the weights, using a zero-mean Gaussian with variance equal to $1/n_l$. Moving one step further He et al. (2015) proposed an initialization with a zero-mean Gaussian with variance equal to $2/n_l$, taking into account the characteristics of the ReLU activation function. Either the input or the output dimension of each layer can be used for $n_l$, as either choice leads to similar results theoretically and experimentally.

In this work, we connect the proposed initialization of the weights with the singular values of the weight matrices and consequently with oversmoothing. For this, we will use the circular law conjecture, which was proven with strong convergence by Tao & Vu (2008).

**Theorem 2 (Circular Law Conjecture)** *Let $N_n$ be a random matrix of order n, whose entries are i.i.d. samples of a zero-mean and bounded variance $\sigma_{std}^2$ random variable. Also let $\lambda_1, ..., \lambda_n$ be the eigenvalues of $\frac{1}{\sigma_{std}\sqrt{n}}N_n$. The circular law states that the distribution of $\lambda_i$ converges to a uniform distribution over the unit disk as n tends to infinity.*

The circular law conjecture dictates the relationship between the standard deviation ($\sigma_{std}$) of the random variable and the radius of the disk. In fact, if we increase $\sigma_{std}$ there is a proportional increment to that particular radius, which in turn increases the largest eigenvalue of $N_n$.

## 3 THEORETICAL ANALYSIS

Despite their success in CNNs and FFNs, existing weight initialization methods fail to capture the effect of the graph topology, which is of high importance in GNNs. Therefore, we generalize the method developed in He et al. (2015) to provide a new weight initialization (G-Init), which takes into account the underlying graph topology.

### 3.1 FORWARD PROPAGATION

In order to simplify the notation, we will use the augmented normalized adjacency matrix (i.e., $\hat{A} = \hat{D}^{-1}(A+I)$) instead of the symmetrically normalized augmented adjacency matrix in Equation 1. This simplification yields the same analysis, except the use of the factor $\frac{1}{d_i}$ instead of $\frac{1}{\sqrt{d_i d_j}}$, in the formula of node representations. Hence, the representation of a node $i$ in layer $l$ becomes:

$$y_l^{(i)} = \frac{1}{d_i} \sum_{j \in \hat{N}(i)} x_l^{(j)T} W^{(l)} + b_l \qquad (2)$$

where $N(i)$ is the neighborhood of node $i$ and $\hat{N}(i)$ is the augmented neighborhood, after self-loop addition and $b_l$ is the bias. Here $x_l^{(j)}$ is an $n_l \times 1$ vector that contains the representation of node $j$ and $W^{(l)}$ is an $n_l \times n_l$ matrix, where $n_l$ is the dimensionality of the layer $l$. Finally, $x_l^{(j)} = \sigma\left(y_{l-1}^{(j)}\right)$, according to Equation 1. Setting $x_l^{(i)'} = \frac{1}{d_i} \sum_{j \in \hat{N}(i)} x_l^{(j)T}$ aligns Equation 2 above with Equation 5 in He et al. (2015).

We let the initial elements of $W^{(l)}$ be drawn independently from the same distribution. Aligned with He et al. (2015), we assume that the elements of $x_l^{(j)}$ are also mutually independent and drawn from the same distribution. Finally we assume that $x_l^{(j)}$ and $W^{(l)}$ are independent of each other. Following a similar analysis as the one presented in He et al. (2015) we get:

$$Var\left[y_l^{(i)}\right] = n_l Var\left[w_l x_l^{(i)'}\right] \qquad (3)$$

where $y_l^{(i)}, x_l^{(i)'}$ and $w_l$ represent the random variables of each element in the respective matrices (Equation 3 above is directly aligned with Equation 6 in He et al. (2015)). We let $w_l$ have zero-mean, leading the variance of the product to be:

$$Var\left[y_l^{(i)}\right] = n_l Var[w_l] E\left[\left(x_l^{(i)'}\right)^2\right] \tag{4}$$

The last equation differs from the corresponding Equation 7 in He et al. (2015), due to the special form of $x_l^{(i)'}$. In fact, $x_l^{(i)'}$ draws information from the neighborhood of each node and combines it through an average operation. Generalizing the work of He et al. (2015) to graph data led to equations, which need special transformations in order to further proceed the analysis. Considering that $x_l^{(i)} = \sigma\left(y_{l-1}^{(i)}\right)$, there is a need to split $x_l^{(i)'}$ into two components, one containing $x_l^{(i)}$ and the other containing the rest information of $x_l^{(i)'}$. In order to achieve that, we will employ the Cauchy–Bunyakovsky–Schwarz inequality (CBS), because it allows to transform a squared sum of elements into a sum of squares of these elements.

**Lemma 1**

$$E\left[\left(x_l^{(i)'}\right)^2\right] = E\left[\frac{1}{d_i^2}\left(\sum_{j\in\hat{N}(i)} x_l^{(j)T}\right)^2\right] \overset{CBS}{\le} \frac{1}{d_i^2}E\left[d_i\sum_{j\in\hat{N}(i)}\left(x_l^{(j)T}\right)^2\right] =$$

$$= \frac{1}{d_i}\left(E\left[\left(x_l^{(i)}\right)^2\right] + k_l^{(i)}\right)$$

where $k_l^{(i)} = \sum_{j\in N(i)}\left(x_l^{(j)T}\right)^2$, i.e. sum of neighbors representations excluding self representation.

Using Lemma 1, Equation 4 transforms to:

$$Var\left[y_l^{(i)}\right] \le \frac{n_l}{d_i}Var[w_l]\left(E\left[\left(x_l^{(i)}\right)^2\right] + k_l^{(i)}\right) \tag{5}$$

If we let $w_{l-1}$ have a symmetric distribution around zero and $b_{l-1} = 0$ then $y_{l-1}$ has zero-mean and symmetric distribution around zero. This leads to $E\left[\left(x_l^{(i)}\right)^2\right] = \frac{1}{2}Var[y_{l-1}]$, when the activation function is ReLU (same analysis as in He et al. (2015)). Applying that result to Inequality 5 yields:

$$Var\left[y_l^{(i)}\right] \le \frac{n_l}{d_i}Var[w_l]\left(\frac{1}{2}Var\left[y_{l-1}^{(i)}\right] + k_l^{(i)}\right) \tag{6}$$

Since we would like to control the variance at the final layer ($L$) of the model, we telescopically replace the factor $Var\left[y_{l-1}^{(i)}\right]$ to Inequality 6 until we reach to $Var\left[y_1^{(i)}\right]$, which is the variance of the first layer of the model.

$$Var\left[y_L^{(i)}\right] \le \frac{n_L}{d_i}Var[w_L]\left(\frac{1}{2}Var\left[y_{L-1}^{(i)}\right] + k_L^{(i)}\right) \implies$$

$$Var\left[y_L^{(i)}\right] \le \frac{n_L}{d_i}Var[w_L]\left(\frac{1}{2}\left(\frac{n_{L-1}}{d_i}Var[w_{L-1}]\left(\frac{1}{2}Var\left[y_{L-2}^{(i)}\right] + k_{L-1}^{(i)}\right)\right) + k_L^{(i)}\right) \implies$$

$$Var\left[y_L^{(i)}\right] \le Var\left[y_1^{(i)}\right]\left(\prod_{l=2}^{L}\frac{n_l}{2d_i}Var[w_l]\right) + \sum_{l=2}^{L}\left(\prod_{j=l+1}^{L}\frac{n_j}{2d_i}Var[w_j]\right)\frac{n_l}{d_i}k_l^{(i)}Var[w_l] \tag{7}$$

These products are the keys to the initialization design, similarly as in He et al. (2015). A proper initialization method should avoid reducing or magnifying the magnitudes of input signals exponentially. We aim to control the upper bound of the variance of the final layer of the model, which in

turn requires tuning of these products in order to take a proper scalar, i.e. 1. A sufficient condition to achieve this, similar as in He et al. (2015), is the following:

$$\frac{n_l}{2d_i}Var[w_l] = 1, \qquad \forall l \tag{8}$$

This leads to a zero-mean Gaussian distribution of the weights, with standard deviation (std) equal to $\sqrt{2d_i/n_l}$.

Our analysis boils down to He et al. (2015), if there is no underlying graph topology. Setting $d_i = 1$ (only self loops) i.e., isolated nodes, results in $k_l^{(i)} = 0$, due to the lack of neighbors and inequality 7 turns into an equality (i.e., Equation 9 in He et al. (2015)).

### 3.2 BACKWARD PROPAGATION

For back-propagation, the gradient for node $i$ is computed by:

$$\Delta x_l^{(i)} = W_l \Delta y_l^{(i)} \tag{9}$$

We follow the same notation as in He et al. (2015), where $\Delta x$ and $\Delta y$ denote gradients $\left(\frac{\partial E}{\partial x} \text{ and } \frac{\partial E}{\partial y}\right)$ and $\Delta y$ is a $n_l \times 1$ vector. To simplify the notation and considering the general trend of GNNs to maintain a constant hidden dimension across layers, we proceed our analysis with $W_l$ being an $n_l \times n_l$ matrix. We also set $\Delta x_l^{(i)'} = \frac{1}{d_i} \sum_{j \in \hat{N}(i)} \left(\Delta x_l^{(j)}\right)$, the average gradient reaching node $i$ based on the forward pass and the interaction with its neighbors (message passing). $\Delta x$ is a $n_l \times 1$ vector representing the gradient.

If we assume that $w_l$ and $\Delta y_l^{(i)}$ are independent of each other, then $\Delta x_l^{(i)}$ has zero-mean for all $l$, when $w_l$ is initialized by a symmetric distribution around zero. Following the above result we also assume that $\Delta x_l^{(i)'}$ has zero-mean.

In back-propagation we have $\Delta y_l^{(i)} = f'(y_l^{(i)})\Delta x_{l+1}^{(i)'}$, where $f'(\cdot)$ is the derivative of $f(\cdot)$. In the case of ReLU, $f'(\cdot)$ is either one or zero, with equal probabilities. We also assume that $f'(y_l^{(i)})$ and $\Delta x_{l+1}^{(i)'}$ are independent. Consequently we have that $E\left[\Delta y_l^{(i)}\right] = E\left[\Delta x_{l+1}^{(i)'}\right]/2 = 0$ (because of the two branches of the ReLU derivative and the independence between $f'(y_l^{(i)})$ and $\Delta x_{l+1}^{(i)'}$) and also $E\left[\left(\Delta y_l^{(i)}\right)^2\right] = Var\left[\Delta y_l^{(i)}\right] = \frac{1}{2}Var\left[\Delta x_{l+1}^{(i)'}\right]$. Finally, we compute the variance of the gradient in Equation 9 as follows:

$$Var\left[\Delta x_l\right] = n_l Var[w_l] Var\left[\Delta y_l^{(i)}\right] = \frac{1}{2}n_l Var[w_l] Var\left[\Delta x_{l+1}^{(i)'}\right] \tag{10}$$

Following a similar approach as in the forward pass and in Lemma 1 we get:

**Lemma 2**

$$Var\left[\Delta x_{l+1}^{(i)'}\right] = E\left[\left(\Delta x_{l+1}^{(i)'}\right)^2\right] \leq \frac{1}{d_i}\left(E\left[\left(\Delta x_{l+1}^{(i)}\right)^2 + o_{l+1}^{(i)}\right]\right) = \frac{1}{d_i}\left(Var\left[\Delta x_{l+1}^{(i)}\right] + o_{l+1}^{(i)}\right) \tag{11}$$

where $o_{l+1}^{(i)} = \sum_{j \in N(i)} \left(\Delta x_{l+1}^{(j)}\right)^2$, *i.e. sum of gradients originating from the neighbors of the node, excluding self-originating gradient.*

Using Lemma 2 in Equation 10 we get:

$$Var\left[\Delta x_l\right] \leq \frac{n_l}{2d_i}Var[w_l]\left(Var\left[\Delta x_{l+1}^{(i)}\right] + o_{l+1}^{(i)}\right) \tag{12}$$

Equation 12 is similar to Equation 7, hence, we arrive at a similar conclusion regarding the initialization of the network, namely, using weights drawn from a zero-mean Gaussian distribution, whose standard deviation is $\sqrt{2d_i/n_l}$.

Similar to He et al. (2015), it is sufficient to use either initialization alone, as they both avoid reducing or increasing exponentially the magnitudes of both the input signals (flowing forward in the network) and the gradients (flowing backward in the network). Hence, we name G-Init the initialization with a zero-mean Gaussian, whose standard deviation is $\sqrt{2d_i/n_l}$.

We note that, in the general case proposed in He et al. (2015), $W_l$ might be substituted by a matrix $\hat{W}_l$, with $\hat{n}_l \times \hat{n}_l$ dimensions, which can be formed by $W_l$ through reshaping. That modification does not affect our analysis except of the appearance of the factor $\hat{n}_l$ in the results instead of $n_l$. More details about the use of $\hat{W}_l$ can be found in He et al. (2015).

## 4 EXPERIMENTS

In this section we analyse experimentally the proposed initialization. In particular, we show the effect of G-Init on the overall performance of the model and explore deeper architectures, in order to highlight its effect on oversmoothing.

### 4.1 EXPERIMENTAL SETUP

**Datasets:** Aligned to most of the literature, we focus on some well-known benchmarks in the GNN domain: *Cora, CiteSeer, Pubmed* and use the same data splits as in Kipf & Welling (2017), where all the nodes except the ones used for training and validation are used for testing. We also utilize the *Arxiv* dataset Hu et al. (2020) of the OGB suite. Finally, we experiment with *Photo, Computers, Physics* and *CS* datasets following the splitting method presented in Shchur et al. (2018).

**Methods:** We experiment with the proposed architecture of GCN (Kipf & Welling, 2017) under five different weight initialization methods. We compare our method (G-Init) against Xavier initialization (Glorot & Bengio, 2010) and Kaiming initialization (He et al., 2015). We also explore two variants of these methods, i.e. drawing samples from a uniform distribution with predefined limits and drawing samples from a zero-mean Gaussian distribution of predefined standard deviation. We use the notation of $Uniform$ and $Normal$ to denote these two variants.

**Hyperparameters:** For GCN we set the number of hidden units (of each layer) to 128 across all datasets. The learning rate is $10^{-3}$ and we vary the depth between 2 and 64 layers.

**Configuration:** Each experiment is run 10 times and we report the average performance and standard deviations over these runs. We train all models for 200 epochs using Cross Entropy as a loss function.

### 4.2 EXPERIMENTAL RESULTS

As we have previously mentioned setting $d_i = 1$ to G-Init ignores the structure of the graph. On the contrary if we set a large value to $d_i$ yields larger weight element values and unstable training. Consequently, we need to find a proper value for $d_i$, which will balance the above two arguments. The smallest possible degree for a node in a graph, with self loops included, equals to 2 (having a single neighbor). We have empirically found, that values in range $(1, 2]$ achieve high performance. In our experiments we have set $d_i = 2$, excluding the *Arxiv* dataset in which we found that $d_i = 1.6$ further improves the performance. In summary, we propose to initialize the weights of a GCN with a zero-mean Gaussian whose standard deviation (std) is $\sqrt{4/n_l}$.

Inequality 7 provides an upper bound to the variance of the signal flowing forward through the network. Applying the proposed G-Init to every layer of the model makes that upper bound equal to $1 + 2\sum_{l=2}^{L} k_l^{(i)}$, which in turn depends on $k_l^{(i)}$. Using G-Init in every layer of the network, results in superior performance compared to Xavier and Kaiming initializations. However, we have obtained even better results, when using G-Init only in the first 80% of the layers and use Xavier initialization to the rest 20% of the layers, which might be attributed to the effect of the uncontrollable term $k_l^{(i)}$ in Inequality 7.

GCN and models utilizing graph convolution, inherently reduce the variance of node representations (signal of information), due to the repeated use of the Laplacian operator. G-Init allows the model

to maintain a higher variance in the lower layers and avoid the collapse of node representation to a subspace, where they would become indistinguishable.

As shown in Table 1 and Table 2 the merits of using G-Init are twofold. Firstly, compared to the other initialization methods, it allows the underlying model to achieve better performance in almost every combination of dataset and depth. In the minority of cases, where G-Init is not the best performing one, it is closely the second best. The experimental results verify the effect of informed weight initialization in GNNs and show how it boosts their performance compared to standard initialization methods that were devised for CNNs and FFNs.

Additional, our extensive experimentation confirms the relationship between weight initialization and oversmoothing. GCN seems to be prone to oversmoothing, when classical initialization methods are used, even in moderate depths. On the contrary, G-Init reduces substantially that effect and facilitates deeper architectures.

The oversmoothing reduction provided by G-Init might attributed to different initial singular values of the weight matrices, compared to the classical initialization methods (i.e., Glorot & Bengio (2010) and He et al. (2015)). These initial values are important in determining the extend of oversmoothing, although we cannot give guarantees about the final values of the singular values of the weight matrices, which result through training.

Again, we draw inspiration from the work of He et al. (2015), who include the activation functions of the model in their analysis and propose a Gaussian with standard deviation equal to $\sqrt{2}/\sqrt{n_l}$. Based on Theorem 2 we conclude that, the method of He et al. (2015) (i.e., Kaiming initialization) creates weight matrices, whose eigenvalues lay on a disk with radius equal to $\sqrt{2}$, while G-Init does the same on a disk of greater radius (i.e., $\sqrt{2d_i}$).

Another possible explanation for the robustness against oversmoothing is that G-Init creates weight matrices, whose largest element is (with large probability) bigger than the largest element of a weight matrix created by one of the classical weight initialization methods. Considering that $s_1 \geq |a_{i,j}^{max}|$ (i.e., largest singular value of a matrix is greater than its largest element), we can conclude that $s_1^{G-Init} > s_1^{Kaiming}$.

Furthermore, Theorem 1 shows the relationship between the largest singular values of the weight matrices and the oversmoothing effect. G-Init initializes the model with weight matrices having larger maximum singular values than those produced by Kaiming initialization. Although this observation does not give guarantees about the final largest singular values, it points out the effect of G-Init over the initial singular values and possibly its relationship to oversmoothing reduction.

## 5 RELATED WORK

The most popular weight initialization methods are those presented in Glorot & Bengio (2010) and He et al. (2015) and they are used for all types of model. Glorot & Bengio (2010) assumed that there was no activation function in the network, in order to propose their initialization method, while He et al. (2015) focused on CNNs with ReLU activation function.

More recently, some literature has appeared that focuses on GNNs. Jaiswal et al. (2022) proposed an initialization based on an analysis of the gradient flow specifically for GCN, coupled with adaptive rewiring. An alternative was proposed by Han et al. (2023), who utilized a trained MLP (trained using only feature vectors, while ignoring graph structure), to initialise GNN weight matrices (GNN weight matrices have the values of the trained MLP weight matrices). Furthermore, Li et al. (2023) proposed a weight initialization based on the decomposition of the variance of each node over the message propagation paths and further decomposition of the variance of each path. Hence, they follow a different approach than G-Init and arrive to different conclusions. On the contrary, our method aims to generalize the ideas of Kaiming initialization on graph data, while making almost no extra assumptions, than the ones used in He et al. (2015).

Regarding the reduction of oversmoothing, which is an important advantage of G-Init, there is rich recent literature. Xu et al. (2018) introduced the skip connection concept in GNNs, when they proposed Jumping Knowledge Networks (JK-Networks). JK-Networks combine information from lower layers of the model with the information reaching its upper-most layer. APPNP (Klicpera et al., 2019) and GCNII (Chen et al., 2020) introduced residual connections in GNNs, and enabled deep architectures. In a similar direction DropEdge (Rong et al., 2020) modified the network's topology, in order to reduce oversmoothing. Despite these methods, which reduce oversmoothing either with architectural or with structural modification, G-Init reduces oversmoothing without changing

Table 1: Performance comparison of different weight initializations of GCN, in *Cora, CiteSeer, Pubmed, Arxiv*. Average test node classification accuracy (%) for networks of different depth. With **bold** is the best performing model for each depth and each dataset.

| # Layers | Method | Cora | CiteSeer | Pubmed | Arxiv |
|---|---|---|---|---|---|
| | | Accuracy (%) | | | |
| 2 | Xavier Normal | $80.45 \pm 0.6$ | $66.68 \pm 1.0$ | $76.06 \pm 0.2$ | $58.97 \pm 0.5$ |
| | Xavier Uniform | $80.62 \pm 0.8$ | $66.82 \pm 0.7$ | $76.00 \pm 0.2$ | $58.60 \pm 0.8$ |
| | Kaiming Normal | $80.44 \pm 0.5$ | $66.67 \pm 1.0$ | $76.13 \pm 0.3$ | $59.60 \pm 0.4$ |
| | Kaiming Uniform | $80.61 \pm 0.8$ | $\mathbf{66.83 \pm 0.8}$ | $75.93 \pm 0.2$ | $59.58 \pm 0.5$ |
| | G-Init | $\mathbf{80.65 \pm 0.5}$ | $66.52 \pm 0.8$ | $\mathbf{76.37 \pm 0.3}$ | $\mathbf{60.52 \pm 0.5}$ |
| 4 | Xavier Normal | $79.80 \pm 0.6$ | $66.48 \pm 1.2$ | $76.52 \pm 0.3$ | $65.81 \pm 0.3$ |
| | Xavier Uniform | $79.49 \pm 0.6$ | $65.93 \pm 1.2$ | $76.43 \pm 0.6$ | $66.10 \pm 0.2$ |
| | Kaiming Normal | $80.17 \pm 0.7$ | $66.74 \pm 0.9$ | $76.77 \pm 0.3$ | $67.69 \pm 0.2$ |
| | Kaiming Uniform | $80.13 \pm 0.8$ | $66.38 \pm 0.9$ | $76.74 \pm 0.5$ | $67.69 \pm 0.2$ |
| | G-Init | $\mathbf{80.87 \pm 0.6}$ | $\mathbf{67.12 \pm 0.7}$ | $\mathbf{76.99 \pm 0.7}$ | $\mathbf{68.17 \pm 0.3}$ |
| 8 | Xavier Normal | $70.49 \pm 4.3$ | $55.66 \pm 2.7$ | $73.52 \pm 2.8$ | $61.44 \pm 0.7$ |
| | Xavier Uniform | $72.63 \pm 5.4$ | $57.13 \pm 6.6$ | $72.72 \pm 2.9$ | $60.92 \pm 0.7$ |
| | Kaiming Normal | $77.05 \pm 2.6$ | $62.20 \pm 2.7$ | $75.14 \pm 2.2$ | $66.97 \pm 0.5$ |
| | Kaiming Uniform | $77.51 \pm 3.0$ | $62.79 \pm 2.5$ | $74.54 \pm 2.3$ | $66.93 \pm 0.4$ |
| | G-Init | $\mathbf{77.77 \pm 1.3}$ | $\mathbf{65.06 \pm 2.2}$ | $\mathbf{75.45 \pm 1.7}$ | $\mathbf{68.04 \pm 0.3}$ |
| 16 | Xavier Normal | $42.74 \pm 10.8$ | $30.26 \pm 9.0$ | $63.58 \pm 15.5$ | $51.11 \pm 2.2$ |
| | Xavier Uniform | $49.42 \pm 7.7$ | $29.71 \pm 14.4$ | $45.11 \pm 16.0$ | $52.50 \pm 2.8$ |
| | Kaiming Normal | $69.72 \pm 4.2$ | $41.64 \pm 12.1$ | $\mathbf{76.02 \pm 1.3}$ | $62.96 \pm 1.1$ |
| | Kaiming Uniform | $72.60 \pm 2.6$ | $41.31 \pm 14.0$ | $75.17 \pm 1.7$ | $62.72 \pm 0.6$ |
| | G-Init | $\mathbf{74.48 \pm 2.1}$ | $\mathbf{56.76 \pm 3.3}$ | $75.33 \pm 1.9$ | $\mathbf{64.83 \pm 0.9}$ |
| 32 | Xavier Normal | $28.07 \pm 6.1$ | $19.73 \pm 7.9$ | $36.99 \pm 12.9$ | $44.31 \pm 2.6$ |
| | Xavier Uniform | $27.24 \pm 3.4$ | $22.45 \pm 8.5$ | $44.24 \pm 9.1$ | $42.63 \pm 2.6$ |
| | Kaiming Normal | $36.60 \pm 11.6$ | $21.68 \pm 7.6$ | $50.80 \pm 9.0$ | $48.84 \pm 4.2$ |
| | Kaiming Uniform | $39.38 \pm 11.6$ | $22.07 \pm 7.9$ | $46.00 \pm 10.2$ | $48.86 \pm 5.3$ |
| | G-Init | $\mathbf{71.66 \pm 2.3}$ | $\mathbf{49.79 \pm 3.9}$ | $\mathbf{75.58 \pm 2.3}$ | $\mathbf{54.72 \pm 2.6}$ |
| 64 | Xavier Normal | $13.01 \pm 7.0$ | $17.77 \pm 2.0$ | $31.74 \pm 11.2$ | $21.55 \pm 6.3$ |
| | Xavier Uniform | $16.25 \pm 8.1$ | $18.51 \pm 2.4$ | $40.13 \pm 3.1$ | $21.27 \pm 6.5$ |
| | Kaiming Normal | $26.61 \pm 8.6$ | $25.16 \pm 5.4$ | $40.52 \pm 11.4$ | $38.08 \pm 6.0$ |
| | Kaiming Uniform | $23.57 \pm 9.4$ | $27.67 \pm 5.0$ | $38.99 \pm 13.9$ | $36.29 \pm 4.4$ |
| | G-Init | $\mathbf{66.30 \pm 5.6}$ | $\mathbf{46.90 \pm 2.7}$ | $\mathbf{72.57 \pm 6.0}$ | $\mathbf{42.12 \pm 3.2}$ |

any of the properties of the network or the graph.

A part of the related work focuses on a better weight initialization of GNNs, while the other focuses on reducing oversmoothing. Our analysis lays in the intersection of these lines of work, presenting a new weight initialization method and showing its effect to oversmoothing reduction.

Table 2: Performance comparison of different weight initializations of GCN, in *Photo, Computers, Physics, CS*. Average test node classification accuracy (%) for networks of different depth. With **bold** is the best performing model for each depth and each dataset.

| # Layers | Method | Accuracy (%) | | | |
|---|---|---|---|---|---|
| | | Photo | Computers | Physics | CS |
| 2 | Xavier Normal | $66.67 \pm 2.8$ | $62.70 \pm 1.3$ | $94.03 \pm 0.1$ | $91.62 \pm 0.4$ |
| | Xavier Uniform | $68.74 \pm 1.6$ | $61.91 \pm 1.2$ | $94.02 \pm 0.0$ | $91.88 \pm 0.2$ |
| | Kaiming Normal | $67.07 \pm 2.9$ | $62.78 \pm 1.4$ | $\mathbf{94.04 \pm 0.1}$ | $91.64 \pm 0.3$ |
| | Kaiming Uniform | $69.10 \pm 1.6$ | $61.98 \pm 1.1$ | $94.02 \pm 0.0$ | $\mathbf{91.89 \pm 0.2}$ |
| | G-Init | $\mathbf{74.37 \pm 2.5}$ | $\mathbf{64.19 \pm 1.2}$ | $94.00 \pm 0.1$ | $91.71 \pm 0.3$ |
| 4 | Xavier Normal | $88.24 \pm 0.5$ | $79.17 \pm 0.8$ | $93.26 \pm 0.1$ | $87.98 \pm 0.5$ |
| | Xavier Uniform | $88.05 \pm 0.9$ | $78.35 \pm 1.5$ | $93.26 \pm 0.1$ | $87.93 \pm 0.4$ |
| | Kaiming Normal | $89.03 \pm 0.8$ | $79.43 \pm 0.7$ | $93.27 \pm 0.1$ | $88.49 \pm 0.4$ |
| | Kaiming Uniform | $88.48 \pm 1.1$ | $78.72 \pm 1.1$ | $93.28 \pm 0.0$ | $88.55 \pm 0.3$ |
| | G-Init | $\mathbf{90.44 \pm 0.3}$ | $\mathbf{79.83 \pm 0.5}$ | $\mathbf{93.28 \pm 0.1}$ | $\mathbf{89.18 \pm 0.3}$ |
| 8 | Xavier Normal | $75.82 \pm 11.7$ | $61.45 \pm 13.0$ | $91.37 \pm 0.5$ | $76.60 \pm 4.2$ |
| | Xavier Uniform | $71.48 \pm 10.3$ | $65.07 \pm 4.8$ | $91.07 \pm 0.8$ | $77.93 \pm 2.3$ |
| | Kaiming Normal | $81.86 \pm 2.1$ | $70.47 \pm 3.6$ | $91.86 \pm 0.2$ | $81.35 \pm 2.2$ |
| | Kaiming Uniform | $83.18 \pm 2.5$ | $68.05 \pm 5.9$ | $91.65 \pm 0.7$ | $82.40 \pm 1.3$ |
| | G-Init | $\mathbf{84.54 \pm 2.0}$ | $\mathbf{74.69 \pm 2.2}$ | $\mathbf{92.00 \pm 0.3}$ | $\mathbf{82.98 \pm 1.6}$ |
| 16 | Xavier Normal | $36.37 \pm 13.4$ | $41.70 \pm 3.1$ | $83.36 \pm 8.0$ | $45.29 \pm 12.5$ |
| | Xavier Uniform | $39.88 \pm 18.4$ | $35.10 \pm 16.6$ | $83.87 \pm 12.7$ | $49.42 \pm 19.7$ |
| | Kaiming Normal | $62.66 \pm 7.3$ | $43.69 \pm 15.7$ | $89.19 \pm 1.8$ | $66.33 \pm 5.0$ |
| | Kaiming Uniform | $60.79 \pm 11.5$ | $48.98 \pm 10.0$ | $87.30 \pm 7.4$ | $62.39 \pm 6.1$ |
| | G-Init | $\mathbf{77.64 \pm 6.1}$ | $\mathbf{65.67 \pm 8.9}$ | $\mathbf{90.80 \pm 0.2}$ | $\mathbf{68.64 \pm 8.1}$ |
| 32 | Xavier Normal | $18.62 \pm 10.7$ | $28.81 \pm 17.5$ | $65.91 \pm 7.0$ | $20.30 \pm 8.3$ |
| | Xavier Uniform | $19.50 \pm 11.9$ | $20.58 \pm 18.3$ | $64.74 \pm 5.1$ | $16.47 \pm 8.8$ |
| | Kaiming Normal | $22.14 \pm 11.0$ | $39.69 \pm 4.0$ | $70.75 \pm 10.0$ | $32.65 \pm 8.9$ |
| | Kaiming Uniform | $31.21 \pm 8.6$ | $24.39 \pm 18.4$ | $76.64 \pm 7.7$ | $28.04 \pm 9.1$ |
| | G-Init | $\mathbf{54.98 \pm 14.7}$ | $\mathbf{40.51 \pm 2.2}$ | $\mathbf{86.15 \pm 6.3}$ | $\mathbf{59.06 \pm 9.5}$ |
| 64 | Xavier Normal | $13.86 \pm 7.5$ | $6.55 \pm 4.7$ | $15.51 \pm 12.2$ | $6.69 \pm 6.2$ |
| | Xavier Uniform | $11.26 \pm 4.4$ | $14.33 \pm 12.7$ | $20.41 \pm 15.6$ | $6.89 \pm 6.6$ |
| | Kaiming Normal | $8.93 \pm 6.4$ | $21.56 \pm 17.4$ | $50.51 \pm 4.1$ | $8.44 \pm 6.1$ |
| | Kaiming Uniform | $9.77 \pm 6.7$ | $17.26 \pm 17.4$ | $50.38 \pm 4.1$ | $8.66 \pm 5.8$ |
| | G-Init | $\mathbf{31.03 \pm 11.7}$ | $\mathbf{38.54 \pm 1.5}$ | $\mathbf{50.78 \pm 0.1}$ | $\mathbf{57.21 \pm 11.0}$ |

## 6 CONCLUSION

We have presented a weight initialisation method that generalizes that of He et al. (2015) to GNNs. We have shown theoretically that the original method is not sufficient for the initialization of GNNs, because it disregards the effect of the underlying graph topology. The new weight initialization scheme is motivated by our theoretical analysis and avoids exponentially large or small values of variance. The method has also been assessed experimentally across a large variety of benchmark datasets. Through these experiments, we have established the relationship between weight initialization and oversmoothing reduction, which allowed us to use deep networks, without modifying either the model's architecture or the graph topology. As a future work we would like to extend G-Init to GNNs that utilize skip and/or residual connections, because such models appear to have better performance than simpler architectures, such as GCN. Finally, we would like to investigate whether G-Init should be adaptive to the depth and how depth or other graph properties affect the choice of $d_i$.

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
