# OpenReview forum: "Informed weight initialization of Graph Neural Networks and its effect on Oversmoothing"
_ICLR.cc/2024/Conference — Submitted to ICLR 2024_

### Official Review · Reviewer_fihQ · 2023-10-18

**Soundness:** 1 poor
**Presentation:** 2 fair
**Contribution:** 1 poor
**Rating:** 3
**Confidence:** 4

**Summary:**

In this paper, the authors propose G-Init for the weights of GCNs by a straightforward use of the informal analysis established by He et al. 2015, which ignores the effects of activation functions. The key idea of the analysis by He et al. 2015 is to initialize the weights of neural networks to control the variance of the forward propagation of features and also backward propagation. The analysis by He et al. 2015 provides no guarantee of the behavior of the features after the neural network is trained. To understand the behavior of the features learned by neural networks, an analysis of the dynamics of neural network features and weights under stochastic gradient descent (or variants) is needed.

The authors provide a few numerical results to show that using G-Init can improve the performance of GCN compared to using a few other weights initialization, including Xavier Normal, Xavier Uniform, Kaiming Normal, and Kaiming Uniform. Then the authors draw the conclusion that G-Init can reduce oversmoothing and enable deep architectures.

Overall, the analysis in this paper is a trivial (near-trivial at best) application of He et al. 2015's results. As far as I am aware, the authors only need to consider the weighted average of a few random variables due to the message passing natural of GCN compared to He et al. 2015, and all calculations are straightforward. The authors draw the conclusion of reducing oversmoothing by confusing the concepts of initialization and the final learned features; instead of analyzing the initialization, the authors should consider the features learned by GCNs after training. The experimental results are extremely weak (the reported accuracies are even significantly worse than that reported by Kipf and Welling in their original GCN paper) and cannot draw the conclusion that G-Init reduces over-smoothing.

From the literature side, the authors have not discussed any work on other efforts on GNN initialization. After checking the literature, it seems to me the paper is very close to "https://assets.amazon.science/df/3f/503aeb5e4e96b2209e55e446f041/on-the-initialization-of-graph-neural-networks.pdf", the existing paper, however, is much more solid in theoretical and numerical results. The forward and backward variance analysis is also analyzed in the existing ICML 2023 paper.

**Strengths:**

Considering weight initialization to improve the performance of GCN is an interesting problem. But it seems to me that has been explored by the existing paper "https://assets.amazon.science/df/3f/503aeb5e4e96b2209e55e446f041/on-the-initialization-of-graph-neural-networks.pdf"

**Weaknesses:**

**The paper seems to be not new** compared to the paper "https://assets.amazon.science/df/3f/503aeb5e4e96b2209e55e446f041/on-the-initialization-of-graph-neural-networks.pdf". If the authors cannot identify the significant novelty over the existing paper. **I would argue for a strong reject.**


**Weaknesses**

1. Both theory and numerical results are rather weak. There is no formal theory, the discussion is a simple use of the discussions in He et al. 2015, which ignores the activation functions.

2. Theorem 1 (Oono & Suzuki) does not indicate oversmoothing yet. The theory in Oono & Suzuki shows that over-smoothing occurs for GCN using the ReLU activation function. Their result is obtained in two steps: First, the spectral radius of $W^{(l)}\otimes \hat{A}$ is no larger than one, which is true since the spectral radius of $\hat{A}$ is no larger than one and $\|W^{(l)}\|$ is small - especially weight decay is used in training GCNs. Second, ReLU reduces the distance of node features to the over-smoothing subspace. Cai & Wang extends the results of Oono & Suzuki to both ReLU and leaky ReLU by analyzing the Dirichlet energy of the node features. Using a different initialization of $W^{(l)}$ cannot reduce oversmoothing; after training using weight decay, over-smoothing is inevitable.

3. The spectral norm of the weights $W$ is small, otherwise, the propagation is unstable for deep architectures. Indeed, if the authors compute the eigenvalue of the weights, the authors will see this. Again, notice that weight decay is used in neural network training as well.

4. Oono & Suzuki provide a toy 2D example that the graph node features converge to the over-smoothing subspace. Perhaps the authors should test that the node features will not converge to that subspace using the new weight initialization.

5. Theorem 2 is a known asymptotic result and the gap between asymptotic and non-asymptotic cases is unknown. As such, it is inappropriate to use this in the finite case. Also, the authors should provide some numerical results to verify the discussion in Section 3, e.g., numerically show the eigenvalues of the weights and plot the evolution of the eigenvalues during training.

6. Showing the slightly improved accuracy does not indicate that oversmoothing does not occur. We cannot even draw the conclusion that oversmoothing happens for GCN in Table 1 and Table 2. If oversmoothing happens, the classification will be uniform for all nodes.

As the authors discussed in the paper, many initialization schemes are proposed to mitigate vanishing gradient. GCN actually suffers from vanishing gradient issues, making deep models perform very badly. GCNII adds skip connections to GCN, and remarkably improves GCN.


7. The reported results for baseline GCN are much weaker than the existing papers, e.g. the original GCN paper, https://arxiv.org/pdf/1609.02907.pdf. I am skeptical about the correctness of the experiments. Can the authors clarify this? For Cora, Citeseer, and Pubmed, the reported accuracy by Kipf and Welling is significantly better than the reported ones in this paper. Moreover, there is no comparison with any other methods or using G-Init in state-of-the-art settings and algorithms, making the value of G-Init unjustifiable.



**Minor**

1. Page 1: "effect of G-Init to …" should be "effect of G-Init on …"

2. Theorem 2 is a known result, and the authors should include a reference similar to that in Theorem 1 to avoid confusion, as this is not a new result established in this paper.

3. The work is only about GCN rather than general GNN, so the authors should use GCN in most cases instead of using GNN.

**Questions:**

See the weaknesses.

---

### Official Review · Reviewer_kvnA · 2023-10-24

**Soundness:** 2 fair
**Presentation:** 2 fair
**Contribution:** 2 fair
**Rating:** 3
**Confidence:** 3

**Summary:**

This work introduces aninitialization scheme called G-Init for Graph Neural Networks (GNNs) to tackle the problem of oversmoothing. GNNs are typically initialized using methods designed for other types of neural networks, which neglect the graph's underlying structure. G-Init considers factors such as signal variance, the impact of graph convolution, both of which contribute to oversmoothing in deep GNNs. Oversmoothing occurs as GNN depth increases and node representations become indistinguishable. The authors demonstrate that G-Init mitigates oversmoothing in deep GNNs and supports the use of deep architectures.

**Strengths:**

1. The paper tackles a very important problem which is the initialization of graph based neural networks.

2. The paper also shows promising results.

**Weaknesses:**

Please refer to questions.

**Questions:**

1. Significant improvements are needed in the writing quality, particularly in Section 3. It would be beneficial if the authors could revise this section.

2. Including supplementary material or background information related to [1] would be a valuable addition.

3. The effectiveness of the method in addressing oversmoothing is unclear. For instance, if all non-linearities were removed, and a deep layer is considered, such as $Z_n = (A^{n}X)W$, the first term, $A^{n}X$, is independent of the initialization. Consequently, oversmoothing may occur even without initialization (of  $A^{n}X$). Unless the initialization can somehow break up the oversmoothening, which I don't see how or why that is possible, without information with regards to the graph, it is hard to intuitively see why initialization can help. Hence, it would be good to understand what is happening from this perspective.

4. Relying solely on test accuracy may not provide a sufficient measure to assess oversmoothing. An example involving a stochastic block matrix with two classes is presented where nodes within each class map to a single representation. In this scenario, oversmoothing may be significant, yet test accuracy remains high. This highlights the need to explore when oversmoothing is genuinely significant and whether the chosen metric is appropriate. It also underscores the importance of investigating in-class and out-of-class oversmoothing. Hence, it would be valuable if the authors could consider alternative metrics like Dirichlet energy to better evaluate the presence of oversmoothing and its connection to test error.

5. The paper's reproducibility raises concerns, as no code is provided, and there is no information about the hyperparameter selection strategy. It is advisable to include this information in the paper.

Based on these concerns, I am inclined to recommend rejecting the paper. However, I am open to revising my evaluation if the authors provide satisfactory responses to these issues.

[1] Kaiming He, Xiangyu Zhang, Shaoqing Ren, and Jian Sun. Delving deep into rectifiers: Surpassing human-level performance on imagenet classification. CoRR, abs/1502.01852, 2015. URL http://arxiv.org/abs/1502.01852.

---

### Official Review · Reviewer_FoM5 · 2023-10-31

**Soundness:** 3 good
**Presentation:** 1 poor
**Contribution:** 1 poor
**Rating:** 3
**Confidence:** 4

**Summary:**

The authors propose G-init, a method to initialize GNNs. The method is based on an extension of the Kaiming initialization, with special treatment in the analysis to account for the graph structure. The authors provide nice theoretical derivations, and then show the behavior of G-init applied to GCN on various datasets.

**Strengths:**

The idea of informed initialization is important for any neural network, and also for graph neural networks, so the paper considers an important point.

**Weaknesses:**

A. The method does not seem novel to me, as it is a slight extension of the Kaiming initialization. Also, as noted by the authors the paper "On the Initialization of Graph Neural Networks" also discusses this aspect of GNNs.

B. The empirical results do not seem to actually avoid oversmoothing. The performance still degrades quite a lot.
Also, the scope of the empirical results are rather narrow, even if the method did not over smooth.

C. Missing literature:
 1.Revisiting graph neural networks: all we have is low pass filters.
  2.A SURVEY ON OVERSMOOTHING IN GRAPH NEURAL NETWORKS

D. The paper is hard to follow and poorly written.

**Questions:**

none

---

### Official Review · Reviewer_8Vyh · 2023-11-02

**Soundness:** 1 poor
**Presentation:** 1 poor
**Contribution:** 2 fair
**Rating:** 1
**Confidence:** 4

**Summary:**

The paper proposes a novel weight initialization scheme for GNNs, addressing the issue of oversmoothing that commonly plagues deep GNNs. The authors provide a theoretical analysis that connects the proposed weight initialization with the singular values of the weight matrices, aiming to prevent the variance of the weights from becoming exponentially large or small.

**Strengths:**

The paper introduces a theoretically informed weight initialization method to mitigate oversmoothness, which is a novel departure from existing approaches.

**Weaknesses:**

The manuscript presents a weight initialization strategy aimed at mitigating the oversmoothing issue.  However, I hold reservations about the utility and impact of this work for the GNN community for the following reasons:

    1). Oversmoothing in GNNs is predominantly attributed to repeated message aggregation. The proposed weight initialization method seems to offer only a marginal solution to this inherent problem. This assertion is supported by the experimental results on the Cora dataset, where the application of the method to a 16-layer model results in a significant drop in accuracy (and for all the other weight initialization methods).

    2). The introduction of additional layers does not appear to yield any accuracy improvements across almost all the datasets evaluated. In fact, the performance detriment is pronounced when the model is extended to 64 layers, where the proposed method exhibits notably poor results. This trend raises concerns about the method's effectiveness, especially as we consider even deeper networks, such as those with 128 or 256 layers, where the challenges may be further amplified.

    3). The scope of the experimental evaluation is somewhat narrow. Notably, there is an absence of experiments on larger datasets, such as those from the OGB.

    4). The clarity of the paper is compromised by suboptimal writing. Several notations are introduced without adequate explanation, which hinders comprehension and detracts from the overall quality of the manuscript.

In light of these points, the paper, in its current form, may not significantly advance the field of GNNs or contribute meaningfully to the ongoing discourse on overcoming oversmoothing.

**Questions:**

See the issues raised in the weaknesses section.

Minor:

- Citation style: please keep consistent.

- Grammatical Oversights: The paper is marred by numerous grammatical errors, particularly concerning punctuation. A glaring oversight is the absence of punctuation marks following ALL equations throughout the document.

---

### Meta-Review · Area_Chair_b77J · 2023-12-12

**Metareview:**

Four referees recommend reject and provide extensive feedback. There was no author response to the initial reviews. Thus I must reject this submission.

**Justification For Why Not Higher Score:**

All referees recommend reject. Authors did not respond to the initial reviews.

**Justification For Why Not Lower Score:**

NA

---

### Decision · Program_Chairs · 2024-01-16

Reject